# Lead-Free AE Sensor Based on BZT–BCT Ceramics

**DOI:** 10.3390/s21217100

**Published:** 2021-10-26

**Authors:** Dong-Jin Shin, Woo-Seok Kang, Dong-Hwan Lim, Bo-Kun Koo, Min-Soo Kim, Soon-Jong Jeong, In-Sung Kim

**Affiliations:** 1Energy Conversion Research Center, Korea Electrotechnology Research Institute (KERI), Changwon 51543, Korea; wlsqkrtk@keri.re.kr (D.-J.S.); dhlim@keri.re.kr (D.-H.L.); bpsh9@keri.re.kr (B.-K.K.); minsoo@keri.re.kr (M.-S.K.); sjjeong@keri.re.kr (S.-J.J.); 2School of Materials Science and Engineering, Ulsan National Institute of Science and Technology (UNIST), Ulsan 44919, Korea; kwsasd84@unist.ac.kr

**Keywords:** lead-free ceramics, AE sensor, piezoelectric sensor

## Abstract

In this study, an acoustic emission (AE) sensor was fabricated using lead-free Ba(Zr_0.2_Ti_0.8_)O_3_–0.5(Ba_0.7_Ca_0.3_)TiO_3_ (BZT–BCT) ceramics. The acoustic and electromechanical properties of the AE sensor were determined by the shapes of the piezoelectric ceramics. To optimize the AE sensor performance, the shapes of the ceramics were designed according to various diameter/thickness ratios (D/T) = 0.5, 1.0, 1.5, 2.0, 2.5, 3.0. The BZT–BCT ceramic with D/T = 1.0 exhibited excellent values of a piezoelectric charge coefficient (*d*_33_), piezoelectric voltage coefficient (*g_33_*), and electromechanical coupling factor (*k_p_*), which were 370 (pC/N), 11.3 (10^−3^ Vm/N), and 0.58, respectively. Optimum values of resonant frequency (*f_r_*) = 172.724 (kHz), anti-resonant frequency (*f_a_*) = 196.067 (kHz), and effective electromechanical coupling factor (*k_eff_*) = 0.473 were obtained for the manufactured BZT–BCT ceramic with D/T = 1.0. The maximum sensitivity and frequency of the AE sensor made of the BZT–BCT ceramic with a D/T ratio of 1.0 were 65 dB and 30 kHz, respectively.

## 1. Introduction

Acoustic emission (AE) is a non-destructive testing method [1,2,3] that refers to a phenomenon in which the energy is emitted as an elastic wave when discontinuous, sudden, and transient motion occurs on the surface or inside an elastic body. AE sensors have excellent sensitivity characteristics and detection capabilities regardless of structures or defects [4]. In addition, AE sensors have been studied as a preventive diagnosis method for power facilities because they are capable of real-time diagnosis even when the facility is in operation [5]. Piezoelectric ceramics in AE sensors are the most important elements that influence the frequency characteristics and sensitivity. Pb(Zr, Ti)O_3_ (PZT) based piezoelectric ceramics are used in most commercial AE sensors. However, owing to restrictions on the use of lead, the application of lead-free piezoelectric ceramics in AE sensors is necessary [6,7].

Ba(Zr_0.2_Ti_0.8_)O_3_–0.525(Ba_0.7_Ca_0.3_)TiO_3_ (BZT–BCT) ceramics reported by Liu and Ren in 2009 showed a high piezoelectric charge constant (620 pC/N) that was comparable to that of PZT in the morphotropic phase boundary (MPB) region [8]. They proposed that there is no free energy barrier in the composition of the ideal tricritical point [8]. Therefore, the polarization can be easily rotated by stress and an E–field because of the vanishing polarization anisotropy. Therefore, BZT–BCT ceramics in the MPB region have excellent piezoelectric and dielectric properties [8,9,10,11,12].

A key factor influencing the frequency characteristics and sensitivity in AE sensors is piezoelectric ceramics. The frequency characteristics are determined by the shape of the piezoelectric ceramics, but the displacement and sensitivity are the characteristics of the piezoelectric material. Therefore, the selection of the piezoelectric materials is very important. In addition, since the AE sensor is a method of detecting a signal by converting an external acoustic wave into electrical energy, it is essential to have good piezoelectric properties. Therefore, BZT–BCT ceramics are receiving a lot of attention in many applications such as sensors, actuators, and energy harvesters. In this work, we hope to confirm the potential of BZT–BCT ceramics in the field of AE sensors and to promote research on AE sensors using lead-free piezoelectric ceramics.

In this study, a lead-free AE sensor was fabricated using BZT–BCT ceramics. To optimize the AE sensor performance, the shapes of the ceramics were designed vs. various diameter/thickness ratios (D/T) = 0.5, 1, 1.5, 2, 2.5, 3. The resonant frequency (*f_r_*), anti-resonant frequency (*f_a_*), mechanical quality factor (*Q_m_*), and electromechanical coupling factors (*k_33_, k_p_, k_eff_*) of BZT–BCT ceramics were determined. Resonant, anti-resonant, and displacement were analyzed vs. the D/T ratios through ATILA simulation. Moreover, the voltage generated vs. the D/T ratios was analyzed through ATILA simulation. Based on this, the D/T ratio of the piezoelectric element was optimized, and the AE sensor was fabricated with the BZT–BCT ceramic made with the optimum D/T ratio, which had the best performance. We investigated the piezoelectric properties of the AE sensor and measured its sensitivity.

## 2. Materials and Methods

BZT–BCT ceramics were fabricated using the conventional solid-state reaction method. BaCO_3_ (purity: 99.9%), TiO_2_ (purity: 99.9%), ZrO_2_ (purity: 99.0%), and CaCO_3_ (purity: 99.0%) were employed as starting materials. The starting powders were weighed vs. the stoichiometric ratio and then ball-milled for 24 h with ethanol and zirconia balls. After drying, the dried powders of BZT–BCT ceramics were calcined at 1350 °C for 2 h. The mixtures were ball-milled for 12 h and then dried at 100 °C for 12 h in an electric oven. Samples with circular shapes were formed by adding 5 wt.% polyvinyl alcohol. The green BZT–BCT ceramics were sintered at 1500 °C for 3 h. The sintered BZT–BCT ceramics were processed vs. various D/T ratios (D/T = 0.5, 1, 1.5, 2, 2.5, 3), as shown in Figure 1. A silver paste was applied on both sides of the specimens and cured at 700 °C for 10 min. Through this process, the electrodes of the specimens were formed. The specimens were poled under a direct-current electric field of 5.94 kV/mm for 1 h in silicone oil at 30 °C. The crystalline structures and morphologies of the BZT–BCT ceramic were characterized by X–ray diffraction (XRD) (θ–2θ scans with a Cu–Kα source) and field emission-scanning electron microscopy (FE-SEM). The piezoelectric coefficient (*d_33_*) was measured by a Berlincourt type quasi-static meter. In addition, the dielectric and electrical properties were measured using an impedance analyzer (Agilent 4294A). The properties of the BZT–BCT ceramics with various D/T ratios are listed in Table 1. To measure the sensitivity characteristics of the fabricated AE sensor, a sensitivity measurement system using the transient acoustic wave measurement method (ASTM 1106–86).

## 3. Results and Discussion

Figure 2a,b shows the XRD pattern and FE–SEM image of the BZT–BCT ceramics. The XRD pattern shows a pure perovskite phase with a tetragonal structure without a secondary phase. The FE–SEM image shows a dense microstructure with low porosity and a uniform grain size distribution. The bulk density of the BZT–BCT ceramic is 5.59 g/cm^3^, and the relative density is 98.2%. Therefore, the properties of the BZT–BCT ceramic were expected to be good.

Table 1 shows the resonant frequency (*f_r_*), anti-resonant frequency (_fa_), mechanical quality factor (*Q_m_*), and electromechanical coupling factors (*k_33_, k_p_, k_eff_*) of the BZT–BCT ceramics with various D/T ratios (D/T = 0.5, 1, 1.5, 2, 2.5, 3). The D/T ratios show a trend to decrease the resonant (*f_r_*) and anti-resonant (*f_a_*) frequencies. The best values were obtained when the D/T ratio was 1, with *Q_m_*, *k_33_*, *k_p_*, *k_eff_*, and volume values of 40.2, 0.51, 0.58, 0.47, and 785 mm^3^, respectively. The electromechanical coupling factor, *k*, is an indicator of the effectiveness with which a piezoelectric material converts electrical energy into mechanical energy, or converts mechanical energy into electrical energy [13]. *k_33_* is a factor for the electric field in direction 3 (parallel to the direction in which the ceramic element is polarized) and longitudinal vibrations in direction 3 (ceramic rod) [13]. Another parameter, *k_eff_*, is frequently used to express the effective coupling coefficient of an arbitrary resonator, either at a fundamental resonance or at any overtone [14]. By using Equations (1) and (2), *k_33_* and *k_eff_* can be respectively expressed as follows:(1)$$k_{33}^{2}=\frac{\pi }{2}\frac{f_{r}}{f_{a}}\tan\left(\frac{\pi }{2}\frac{f_{a}-f_{r}}{f_{a}}\right)$$
(2)$$k_{eff}=\sqrt{\frac{f_{a}^{2}-f_{r}^{2}}{f_{a}^{2}}}$$
where *f_r_* is the resonant frequency; *f_a_* is the anti-resonant frequency of the piezoelectric ceramics.

Figure 3 shows the impedance’s magnitude and phase angle vs. the frequency of BZT–BCT ceramics with various D/T ratios. In the simulation, the material properties were completed with the measurable properties of our samples and with the reference [15]. As the D/T ratio decreased, the resonant and anti-resonant frequencies decreased. The resonant and anti-resonant frequencies both decreased as the thickness of the sample increased. This can be explained by the following equation of the resonant frequency of the AE sensor [16].
(3)$$f_{r}=\frac{\upsilon }{2l}=\frac{1}{2l}\sqrt{\frac{c}{\rho }}$$
where l is the length (thickness) of the piezoelectric ceramics; υ is the wave velocity; c is the effective elastic stiffness; and ρ is the mass density on the piezoelectric ceramics. As shown in the above equation, it can be seen that the resonant frequency of the AE sensor decreases as the thickness of the piezoelectric ceramic increases. As shown in Table 1, the D/T ratio of the specimen with the best characteristics was 1. At D/T = 1, the resonant and anti-resonant frequencies were 172.724 and 196.067 kHz.

Figure 4 shows the impedance’s magnitude vs. the frequency and displacement of BZT–BCT ceramics at the resonant frequency vs. the D/T ratio by ATILA simulation. The simulation result on the right is the displacement of BZT–BCT ceramics at the resonant frequency. The simulated and actual measured resonant and anti-resonant frequencies do not exactly match, but trends can be inferred. As the D/T value decreases, the trend to move to the low frequency coincides, and at D/T = 1 and D/T = 1.5, the simulation and measurement results of impedance’s magnitude vs. frequency show very similar trends. It was confirmed that the piezoelectric element with D/T = 1 operates in the combined vibration mode of the radial mode and the thickness direction as shown in the simulation results. As shown in Figure 4, it is considered that the difference between the simulation result and the actual measurement result is due to the difference in the machining accuracy. In addition, the physical properties of the device and the measurement environment in the simulation will be a very ideal and flawless environment. In addition, the physical properties of the device and the measurement environment in the simulation will be a very ideal and flawless environment. However, this is not the case in the actual device measurement environment. Therefore, we assumed the error was caused by the difference between these two environments.

Figure 5 shows the output voltages of BZT–BCT ceramics in the ATILA simulation vs. the D/T ratio. This figure is the result of measuring the output voltage generated in the piezoelectric ceramic when a force of 1 N is applied. In the simulation result, the white dot indicates the point where the voltage is measured, and the output voltage measured at that point is shown below. It can be seen that the generated voltage increases as the D/T ratio decreases. The following equation indicates the amount of the output voltage that the piezoelectric material can generate when an external pressure is applied [17]:(4)$$V=-\left(g_{33}hT\right)$$
where V is the generated output voltage of the piezoelectric ceramics; h is the piezoelectric ceramic thickness; and T is the stress on the piezoelectric ceramics. As shown in the above equation, if the same stress is applied to the same piezoelectric material, the generated voltage is proportional to the thickness of the piezoelectric element. Therefore, as it can be seen from the simulation results, the generated voltage tends to increase as the D/T ratio decreases.

Figure 6a,b shows the piezoelectric charge coefficient (*d_33_*) and voltage coefficient (*g_33_*) of BZT–BCT ceramics vs. the D/T ratio, respectively. As shown in Figure 6a, the piezoelectric charge coefficient (*d_33_*) increased until D/T = 1.5, and then decreased. The BZT–BCT ceramic with a D/T ratio of 1.5 attained the highest piezoelectric charge coefficient of 377 pC/N. In Figure 6b, the piezoelectric voltage coefficient (*g_33_*) tended to decrease when the D/T ratio was higher than 1. The BZT–BCT ceramic with a D/T ratio of 1 attained the highest piezoelectric voltage coefficient (*g_33_*) of 11.38 × 10^−3^ Vm/N. The piezoelectric voltage coefficient indicates the electric field that the piezoelectric material can generate when an external pressure is applied:(5)$$g_{33}=\frac{d_{33}}{\varepsilon_{r}\varepsilon_{0}}$$
where *g_33_* is the piezoelectric voltage coefficient; *ε_r_* is the permittivity of the piezoelectric material; *ε_0_* is the vacuum permittivity; *d_33_* is the piezoelectric charge coefficient (or the piezoelectric strain coefficient).

Figure 7a,b shows the dielectric constant and *FOM_33_* of BZT–BCT ceramics vs. the D/T ratio, respectively. As shown in Figure 7a, the dielectric constants of the BZT–BCT ceramics increased up to D/T = 1.5, and then decreased. The BZT–BCT ceramic with a D/T ratio of 1.5 attained the highest dielectric constant of 3814. The dielectric constant of BZT–BCT ceramics vs. the D/T ratio is the result of measurement after polarization treatment. This is a result calculated by measuring the capacitance at 1 Hz, and it shows a difference vs. the D/T ratio despite the ceramic of the same composition. We presume that this difference arises from the following reasons. After polarization treatment, the dielectric constant is determined by the complementation of a factor increasing the dielectric constant (compressive stress caused by electrostriction or a piezoelectric effect) and a factor decreasing the dielectric constant (arrangement of the dipoles in the polarization direction) [18]. *FOM_33_* effectively describes the change in stored electrical energy within a piezoelectric material when stress is applied. By using Equation (5), *FOM_33_* can be expressed as follows [13]:(6)$$FOM_{33}\left({\mathrm{pm}}^{2}/N\right)=d_{33}\times g_{33}=\frac{{\left(d_{33}\right)}^{2}}{\varepsilon_{r}\varepsilon_{0}}$$

The trend of *FOM_33_* is similar to that of the piezoelectric charge coefficient (*d_33_*). Similar to the other results, the BZT–BCT ceramic with a D/T ratio of 1.5 attained the highest *FOM*_33_ value of 4.21 pm^2^/N.

Figure 8 shows a schematic and an image of the AE sensor. The AE sensor was fabricated with a structure based on a BZT–BCT ceramic with a D/T ratio of 1, which was the best in terms of performance. In addition, a stainless-steel case and case cover were used to implement a grounding structure to block external noise. The wear plate (alumina) performed acoustic impedance matching between the material to be measured and the BZT–BCT ceramic. The upper electrode of the BZT–BCT ceramic was connected to the signal line of the BNC cable and connected to an external amplifier. The lower electrode of the BZT–BCT was connected to the wear plate and the case to form a grounding structure.

Figure 9 shows a schematic of the measurement system of the AE sensor. The pencil lead break (PLB) method is a standardized test method according to the specifications of the American Society for Testing and Materials (ASTM) [18]. This technique was standardized by the ASTM under the designation E976–1018. The pencil lead was broken at a specific angle with respect to the surface of the forged steel, and the resulting elastic wave was measured using an AE sensor. The pencil lead used in the PLB method was only used to meet specific conditions, such as lead thickness (0.5 mm), length (3 mm), and hardness (2 H). We performed the PLB method at specific angles and positions using Nelson shoes to maintain repeatability and accuracy. The PLB was used as the elastic wave source, and forged steel (diameter 40 cm, thickness 5 cm) with less attenuation and dispersion was used as the transmission medium. The AE sensor made of the BZT–BCT ceramic was attached to the left side, 5 cm away from the acoustic wave source with a glycerin contact medium and connected to the computer through an external pre-amplifier. The standard AE sensor (PHYSICAL ACOUSTICS GROUP_R15I–AST) with a built-in pre-amplifier was attached to the right side for signal calibration. The elastic wave generated by the PLB method arrives at the standard sensor and the sensor made of the BZT–BCT ceramic. The signals detected by these two sensors are displayed in the program (PACwin Software) via an AE measurement system (MISTRAS_PCI–2 AE system) connected to a computer. The sensitivity was calculated by the following equation.
(7)$$dB_{ae}=20\times log_{10}\left(\frac{V_{in}}{10^{-6}}\right)-Gain$$
where *V_in_* is the output voltage measured by an AE sensor, and the gain is the value set by the preamplifier connected to the sensor.

Figure 10 shows the detected signals in the time domain from the AE sensor based on the BZT–BCT ceramic. The peak amplitude of the lead-free sensor made of BZT–BCT ceramic was similar to that of the lead-free AE sensor reported in other studies [6,19]. To reduce the noise level and detect a more accurate AE waveform, we increased the noise threshold level to detect the AE signal waveform when a hit occurred. The main peak at the voltage signals of the AE sensor was determined from the surface wave rather than from longitudinal or transverse waves [7]. Because the actual AE sensor also detects several reflected waves occurring at the boundary of the forged steel, several detected waves were measured, which were larger than the number of theoretical velocity waves [7].

Figure 11 shows the sensitivity of the AE sensor fabricated using BZT–BCT. The AE sensor made of the BZT–BCT ceramic had a frequency range of 20–130 kHz when the sensitivity was over 50 dB. As shown in Figure 9, the peak sensitivity frequency and sensitivity of the lead-free AE sensor were 30 kHz and 65 dB, respectively. It can be seen that the lead-free sensors had a sensitivity comparable to that of other lead-free AE sensors [6,20]. In general, the frequency range of the resonant-type sensor was 75–150 kHz, and most of the AE events were likely to occur. In addition, the sensitivity of the AE sensor using the thickness-shear-mode piezoelectric device was remarkably higher than that of the AE sensor using the radial mode and other modes [7,21]. Therefore, it was determined that the AE sensor using lead-free piezoelectric ceramics can be effectively applied to the field of diagnostic technology in power facilities.

## 4. Conclusions

In this study, the lead-free AE sensor was fabricated using BZT–BCT ceramics. To optimize the AE sensor performance, the shapes of the ceramics were designed vs. various D/T ratios = 0.5, 1, 1.5, 2, 2.5, 3. Resonant, anti-resonant, and displacement properties were analyzed vs. the D/T ratio through ATILA simulation. Moreover, the voltage generated vs. the D/T ratio was analyzed through ATILA simulation. Thus, the D/T ratio of the piezoelectric element was optimized, and the AE sensor was fabricated with the BZT–BCT ceramic made with the optimum ratio of D/T to analyze its characteristics.

The bulk density of the BZT–BCT ceramic was 5.59 g/cm^3^, and the relative density was 98.2%.The best values were obtained when the D/T ratio was 1, with *Q_m_*, *k*_33_, *k_p_*, *k_eff_*, and volume values of 40.2, 0.51, 0.58, 0.47, and 785 mm^3^, respectively.The actually fabricated BZT–BCT ceramic tended to have a lower *f_r_* as the D/T ratio decreased.In ATILA simulation, the *f_r_* of the BZT–BCT ceramic tended to decrease as the D/T ratio decreased.In ATILA simulation, when a force of 1 N was applied, as the D/T ratio decreased, the output voltage increased. The highest output voltage value was 152.97 V at the D/T ratio of 0.5.The BZT–BCT ceramic with the D/T ratio of 1.5 attained the highest *d*_33_ of 377 pC/N.The BZT–BCT ceramic with the D/T ratio of 1 attained the highest *g*_33_ of 11.38 × 10^−3^ Vm/N.The BZT–BCT ceramic with the D/T ratio of 1.5 attained the highest *FOM*_33_ value of 4.21 pm^2^/N.An AE sensor was fabricated by the BZT–BCT ceramic with the D/T ratio of 1 showing the best characteristics, and the AE sensor characteristics were evaluated by the PLB method.The peak output voltage of the AE sensor was 0.6 V.The maximum sensitivity and frequency of the AE sensor were 65 dB and 30 kHz, respectively.

Table 2 shows the piezoelectric and AE characteristics of piezoelectric ceramics for AE sensors. The BZT–BCT ceramic has a higher density and better piezoelectric charge coefficient values than other lead-free ceramics. In addition, it can be seen that the electromechanical coupling coefficient is also superior to that of other lead-free piezoelectric ceramics. The sensitivity is excellent enough to be comparable to that of lead based piezoelectric ceramics. After design and optimization, a highly sensitive AE sensor was fabricated to validate the feasibility of the BZT–BCT ceramic for AE sensor applications.

## Figures and Tables

**Figure 1 sensors-21-07100-f001:**
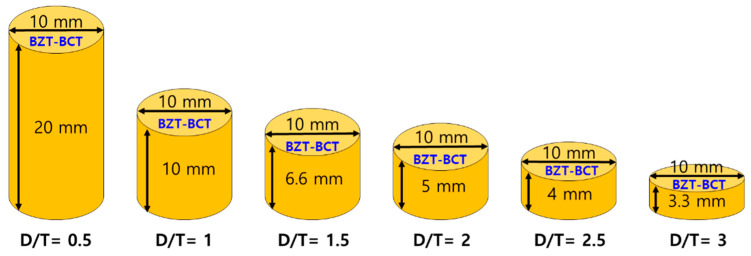
Schematic diagram of BZT–BCT ceramics with various D/T ratios (D/T = 0.5, 1, 1.5, 2, 2.5, 3).

**Figure 2 sensors-21-07100-f002:**
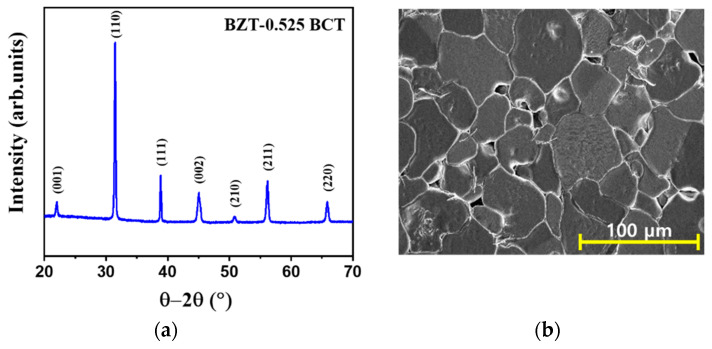
(**a**) XRD pattern and (**b**) FE–SEM image of the BZT–BCT ceramics.

**Figure 3 sensors-21-07100-f003:**
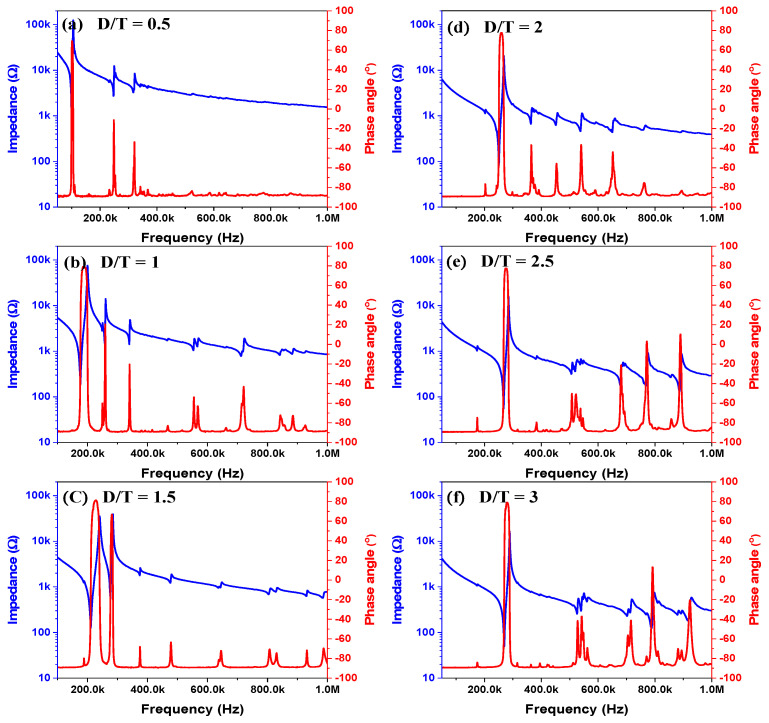
Impedance’s magnitude and phase angle vs. frequency of BZT–BCT ceramics with various D/T ratios (D/T): (**a**) D/T = 0.5, (**b**) D/T = 1, (**c**) D/T = 1.5, (**d**) D/T = 2, (**e**) D/T = 2.5, (**f**) D/T = 3.

**Figure 4 sensors-21-07100-f004:**
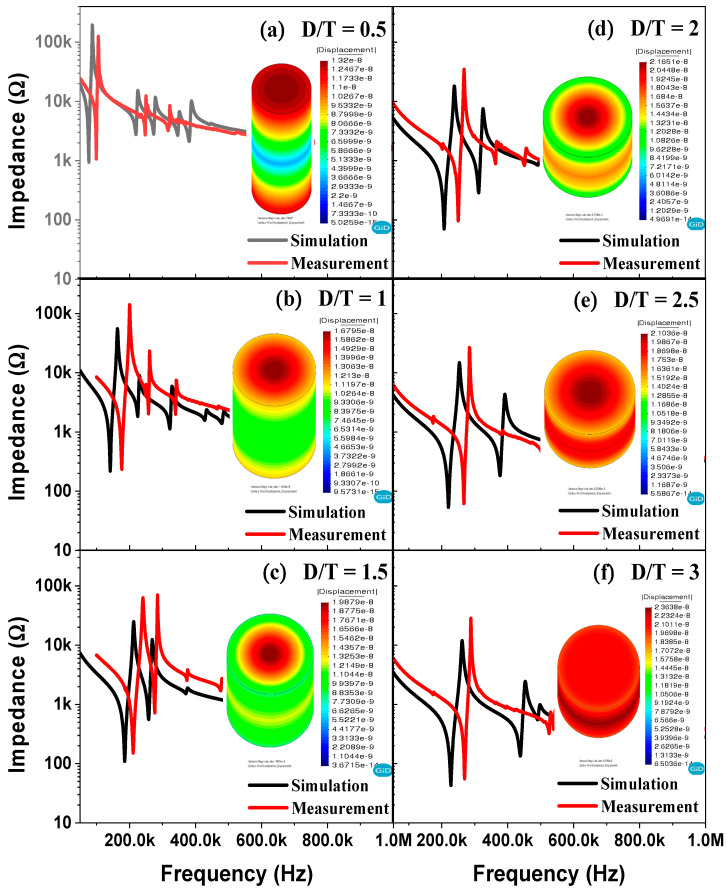
Impedance’s magnitude vs. frequency and displacement of BZT–BCT ceramics at resonant frequency vs. the D/T ratio by ATILA simulation: (**a**) D/T = 0.5, (**b**) D/T = 1, (**c**) D/T = 1.5, (**d**) D/T = 2, (**e**) D/T = 2.5, (**f**) D/T = 3.

**Figure 5 sensors-21-07100-f005:**
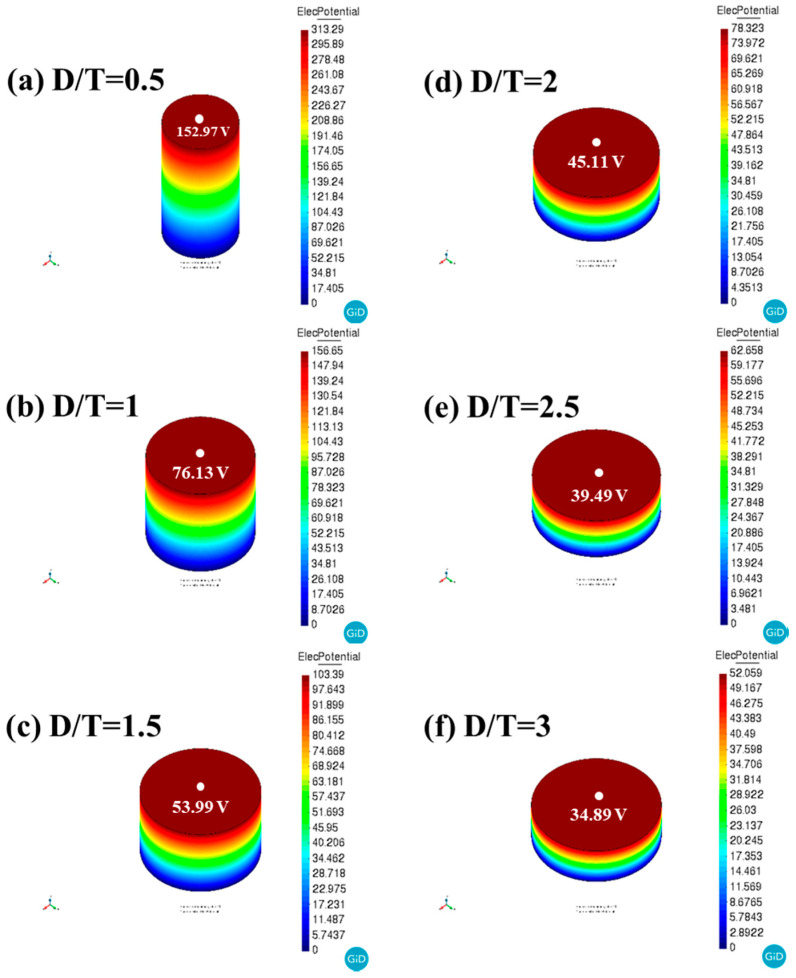
Output voltages of BZT–BCT ceramics in ATILA simulation vs. the D/T ratio.

**Figure 6 sensors-21-07100-f006:**
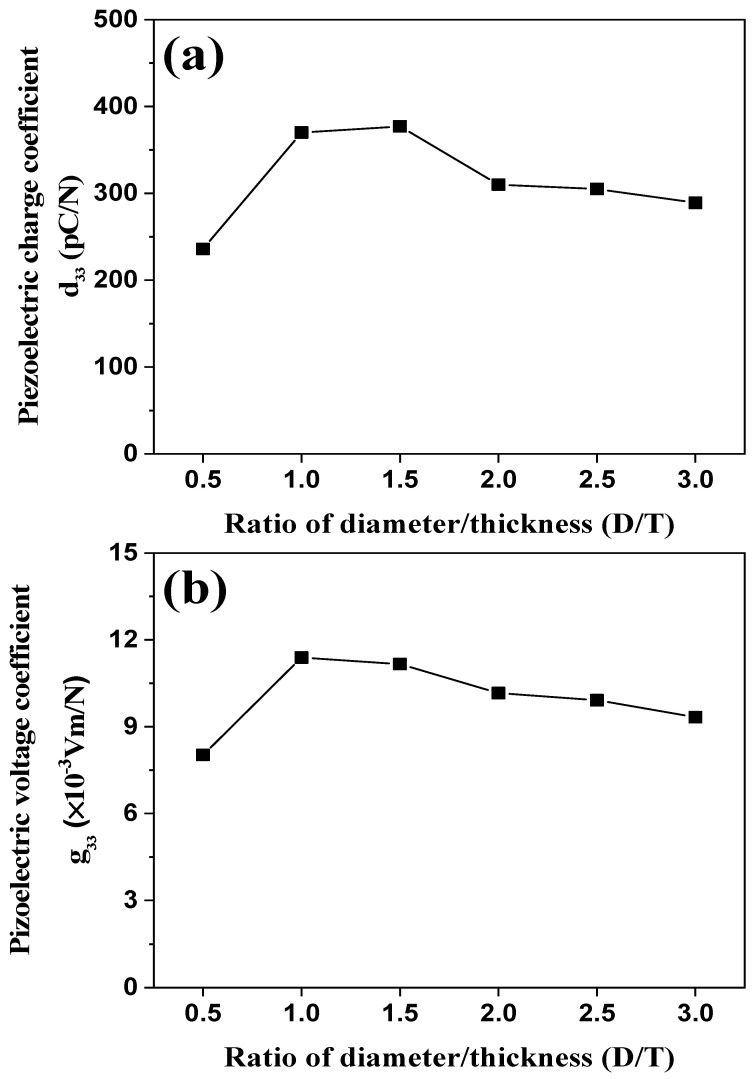
(**a**) Piezoelectric charge coefficient and (**b**) piezoelectric voltage coefficient of BZT–BCT ceramics vs. the D/T ratio.

**Figure 7 sensors-21-07100-f007:**
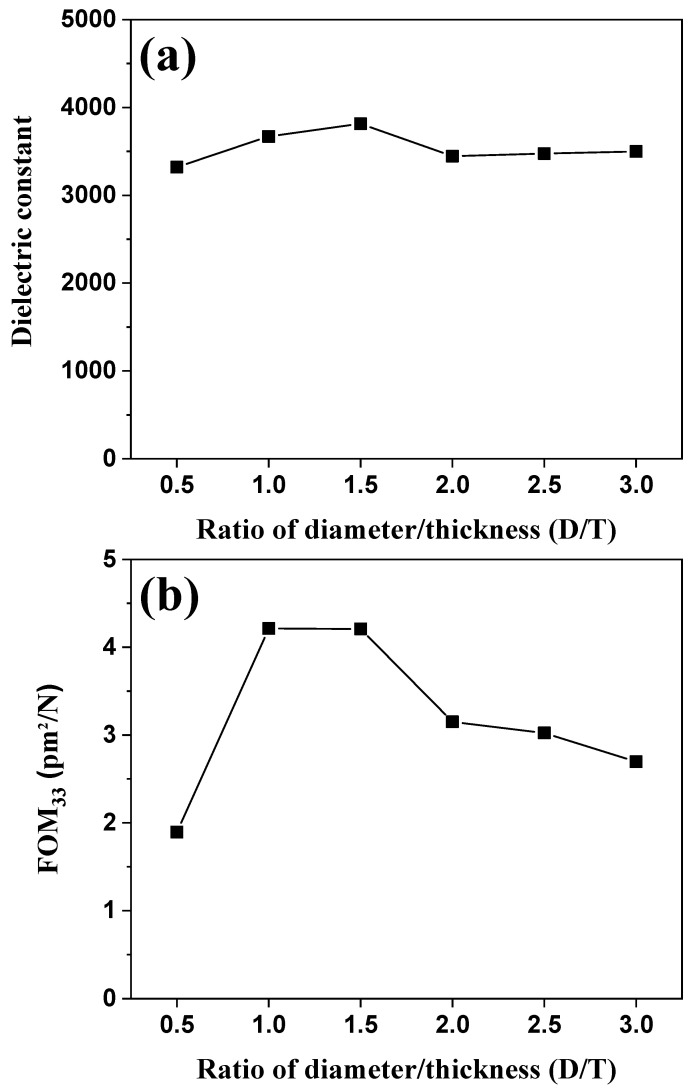
(**a**) Dielectric constant and (**b**) *FOM*_33_ of BZT–BCT ceramics vs. the D/T ratio.

**Figure 8 sensors-21-07100-f008:**
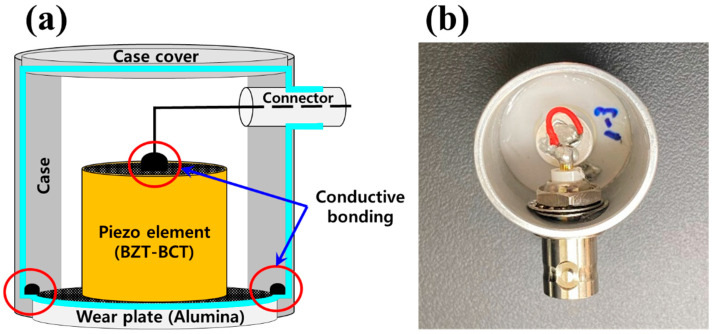
(**a**) Schematic diagram and (**b**) image of the AE sensor.

**Figure 9 sensors-21-07100-f009:**
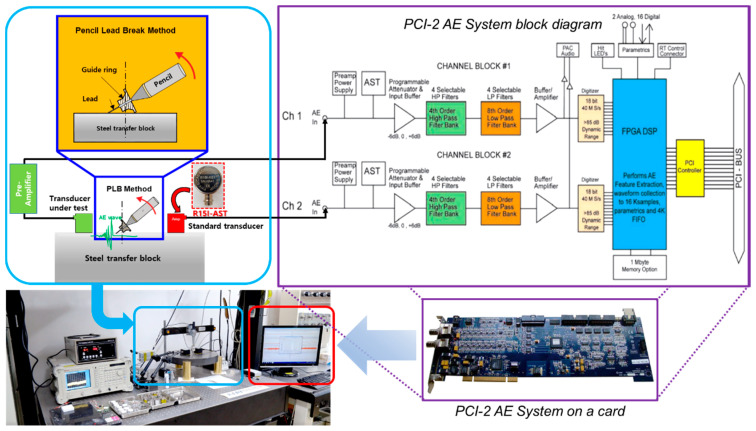
Schematic diagram of measurement system of the AE sensor.

**Figure 10 sensors-21-07100-f010:**
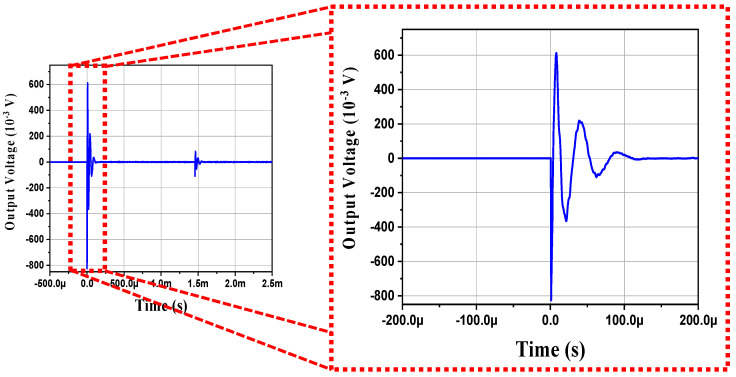
Output voltage of the AE sensor.

**Figure 11 sensors-21-07100-f011:**
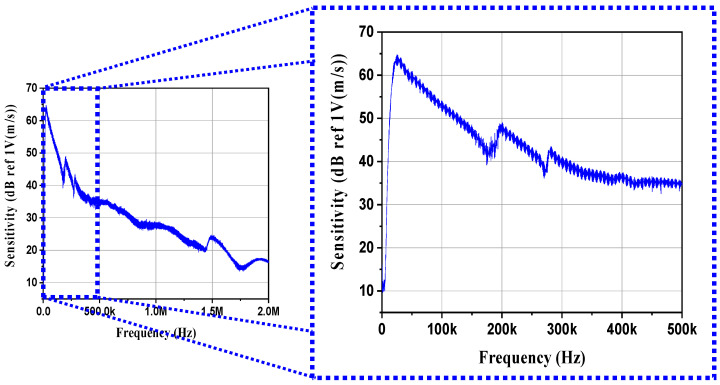
Sensitivity of the AE sensor.

**Table 1 sensors-21-07100-t001:** Resonant frequency (*f_r_*), anti-resonant frequency (*f_a_*), mechanical quality factor (*Q_m_*), and electromechanical coupling factors of the BZT–BCT ceramics with various D/T ratios.

Diameter (D)	Thickness (T)	D/T	*f_r_* (kHz)	*f_a_* (kHz)	*Q_m_*	*k* _33_	*k_p_*	*k_eff_*	Volume (mm^3^)
10	20	0.5	95.033	100.811	87.66	0.36	0.39	0.33	1570
10	10	1	172.724	196.067	40.20	0.51	0.58	0.47	785
10	6.6	1.5	208.225	235.428	42.34	0.50	0.57	0.46	518
10	5	2	246.239	265.751	127.85	0.41	0.44	0.37	392
10	4	2.5	263.901	282.842	373.86	0.39	0.42	0.35	314
10	3.3	3	266.659	287.786	123.57	0.41	0.44	0.37	259

**Table 2 sensors-21-07100-t002:** Piezoelectric and AE sensor characteristics of piezoelectric ceramics for AE sensors.

	Density (kg/m^3^)	*d*_33_ (pC/N)	*k_p_*	*g*_33_ (10^−3^ Vm/N)	*Q_m_*	Sensitivity (dB)
PZT 5H [6]	7600	550	0.62	18.3	70	65
PZT 5A [22]	7500	380	0.62	25.0	100	69
KNN–Cu [6]	4500	75	0.42	33.9	800	57
KNN–LS [23]	4530	261	0.44	40.1	104.9	55
KNN–LTS [22]	4600	300	0.49	27.0	54.76	66
BZT–BCT [This work]	5590	370	0.58	11.3	40.2	65

## Data Availability

Not applicable.

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
