# Peer review of "Lead-Free AE Sensor Based on BZT–BCT Ceramics"

_sensors, 2021, doi:10.3390/s21217100_

Round 1

Reviewer 1 Report

The authors have submitted a manuscript of=n Lead-free AE Sensor based on BZT–BCT Ceramics. I have few comments before this can be considered further. 

There are some spelling mistakes and also the grammar needs to be rechecked. 

Looking at the SEM image, I can see porosity, Authors did mention porosity, however, density should also be provided. 

Table 1 displays the resonant frequency and other piezoelectric parameters, however, authors did not mention the reason for the reason for decreasing resonant frequency with increasing volume. 

Page 4 line 137, please provide a comparison for excellent electromechanical coupling.

Page 5, author stated there is a difference between simulation and measured values, the reason provided is very vague, please provide a clear reason of this difference and explain.

In experimental section, authors mentioned the sintering time was in the range, please explain why is this the case. Also, etyl alcohol is more commonly used as ethanol. 

How did author come up with a poling voltage of 5.9436 kV/mm. Please narrow it down to either 6 or provide a justification why such precise voltage is needed, how was this maintained and effect on the samples. 

Dielectric constant is a materials property and should stay the same. Authors mentioned that by varying the D/T ratio, it changed., how would author justify this phenomenon. 

Conslusion must be improved, it is not an abstract. 

Genrelly speaking, the paper reports data and the key scientific understanding is missing in most part of the paper. 

Reviewer 2 Report

Please find this reviewer's opinion on the manuscript in the Report attached as a PDF file.

Round 2

Reviewer 1 Report

The authors have successfully addressed my comments and am happy with the changes.

Author Response

Your comments have helped me to improve my manuscript more logically and scientifically.

Thanks for your insightful advice.